# COVID-19 Changes Public Awareness about Food Sustainability and Dietary Patterns: A Google Trends Analysis

**DOI:** 10.3390/nu14224898

**Published:** 2022-11-19

**Authors:** Carlos Portugal-Nunes, Liliana Cheng, Mariana Briote, Cristina Saraiva, Fernando M. Nunes, Carla Gonçalves

**Affiliations:** 1CECAV-Veterinary and Animal Science Research Centre, 5001-801 Vila Real, Portugal; 2Faculty of Psychology, Education and Sport, Lusófona University of Porto, 4000-098 Porto, Portugal; 3Biology and Environment Department, School of Life Sciences and Environment, University of Trás-os-Montes e Alto Douro, 5000-801 Vila Real, Portugal; 4Shared Assistential Resources Unit (URAP), Grouping of Health Centers of the Middle Tagus, Regional Health Administration of Lisbon and Tagus Valley, 1700-122 Lisbon, Portugal; 5Department of Veterinary Sciences, School of Agrarian and Veterinary Sciences, University of Trás-os-Montes e Alto Douro, 5000-801 Vila Real, Portugal; 6Food and Wine Chemistry Laboratory, CQ-VR-Chemistry Research Centre-Vila Real, University of Trás-os-Montes and Alto Douro, 5000-801 Vila Real, Portugal; 7Chemistry Department, School of Life Sciences and Environment, University of Trás-os-Montes and Alto Douro, 5000-801 Vila Real, Portugal; 8CITAB-Centre for the Research and Technology of Agro-Environmental and Biological Sciences, University of Trás-os-Montes and Alto Douro, 5000-801 Vila Real, Portugal; 9EPI Unit, Instituto de Saúde Pública, Universidade do Porto, 4150-564 Porto, Portugal

**Keywords:** food sustainability, sustainability, healthy diet, Mediterranean diet, flexitarianism, Google trends

## Abstract

The COVID-19 pandemic has not only affected healthcare systems and global economies but also directly impacted food security and purchasing behaviors. The aim of this study is to investigate if COVID-19 has induced changes in public interest regarding Food Sustainability and healthy-sustainable dietary patterns across Europe and in European regions. A Google Trends search was performed using the search terms “Food Sustainability + Sustainable Diet + Sustainable Food” (grouped as “Food Sustainability”) and the topics “Sustainability”, “Healthy Diet”, “Mediterranean Diet”, and “Flexitarianism” for the years 2010 to 2022. Data were obtained for 12 countries in Europe. The trends in interest after the COVID-19 outbreak were forecast based on previous data. After the COVID-19 outbreak, an increase in Food Sustainability interest was observed and was higher than forecast based on the previous data. A significant interest increase in Sustainability was observed; nevertheless, this increase was smaller than the forecast increase. Mixed results were obtained for dietary patterns across European regions, yet, considering the mean interest for Europe, it seems that the COVID-19 pandemic outbreak dampened the interest in dietary patterns such as the Healthy Diet and Flexitarianism and promoted an interest in the Mediterranean Diet. Understanding consumers’ beliefs and behaviors toward food choices is crucial for the transition towards sustainable diets, and definitions of educational and behavioral interventions are essential to this transition.

## 1. Introduction

The COVID-19 pandemic significantly changed world economy dynamics and brought new concerns for countries [1,2]. To prevent and limit the transmission of the novel coronavirus, confinements and lockdowns were imposed by several governments. The restrictions to individual liberty directly impacted food security, food waste, and purchasing behaviors [3]. In 2020, the Food and Agriculture Organization of the United Nations (FAO), the International Fund for Agricultural Development (IFAD), the World Bank, and the World Food Program (WFP) at the Extraordinary G20 Agriculture Ministers’ Meeting, released a joint statement on the COVID-19 impacts on food security and nutrition, which concluded that the “pandemic is already affecting the entire food system and collective action is needed to ensure that markets are well-functioning” [4]. Taken together, the alterations in consumer behavior and food systems induced by COVID-19 placed a special emphasis on Food Sustainability.

The COVID-19 pandemic added stress to food systems, which were already facing sustainability challenges. In 2019, food systems were responsible for 31% of human-caused greenhouse gas (GHG) emissions [5,6]. In recent years, in the context of population growth, food transition, and climate change, several studies have focused on the environmental impacts of food systems, in addition to diet’s influences on human health [7,8]. Changing the global food system and emphasizing more environmentally sustainable and healthy diets are solutions to achieve the Sustainable Development Goals (SDGs), and the Paris Climate Agreement, as well as other international sustainability targets [9]. Dietary changes are needed to improve public health, nutritional outcomes, and increase the environmental sustainability of the food system [10,11].

In 2019, the Intergovernmental Panel on Climate Change (IPCC), the United Nations body for assessing the science related to climate change, published a special report on climate change and land where they addressed the mitigation potential of different diets [12]. Among the analyzed dietary patterns, we can find the Mediterranean diet (MedDiet), the healthy diet, and the flexitarian dietary pattern. The MedDiet, on top of its health benefits [13], has also been recognized as an example of a sustainable dietary pattern [14,15]. The MedDiet is characterized by the consumption of cereals, fruits, vegetables, legumes, tree nuts, seeds, and olives, with olive oil as the principal source of added fat, along with high to moderate intakes of fish and seafood, the moderate consumption of eggs, poultry, and dairy products (cheese and yogurt), the low consumption of red meat, and a moderate intake of alcohol (mainly wine during meals) [13,16]. Healthy diet is defined in the IPCC report as a pattern based on global dietary guidelines for the consumption of red meat, sugar, fruits and vegetables, and total energy intake. Flexitarianism, one of the dietary patterns with higher mitigation potential, was defined by the substitution of 75% of meat and dairy by cereals and pulses; at least 500 g per day of fruits and vegetables; at least 100 g per day of plant-based protein sources; modest amounts of animal-based proteins; and limited amounts of red meat (one portion per week), refined sugar (less than 5% of total energy), vegetable oils high in saturated fat, and starchy foods with a relatively high glycaemic index. Also, in 2019, considering the diet–environment–health trilemma, the EAT-Lancet report acknowledged the need of a “Planetary healthy reference diet” that in its concept is very similar to a “flexitarian” diet [17,18].

While there is evidence supporting the health, environmental, and economic advantages of the transition to healthy and sustainable dietary patterns, several barriers exist and need to be understood to be overcome. Several factors influence consumers’ food choices and dietary behavior, including individual perceptions, sociocultural factors, food cost, and the existing food environment (including food availability and affordability) and biological determinants (genes). Human behavior affects, and may even drive, the future of global sustainability [19]. Therefore, consumer behavior and decisions may play an important role in the transition to a more sustainable food system [19,20,21]. Consumers, through their choices, in terms of type of product, quantity, and quality, among other factors, direct the sustainability of food systems [22]. The drivers of consumers’ choices may include the social norm effect, self-efficacy (perceived behavioral control), and various dimensions, such as health, environment, economics, society, and culture. Moreover, consumers’ choices are directed by the information that they seek and that is made available to them [23,24]. Understanding public interest in and demand for information on food sustainability and dietary patterns is paramount to understanding the ability and susceptibility of consumers to change towards healthy and sustainable food systems, even more in a time where the sustainability of the food system is challenged [25].

The accessibility of the Internet and the rise of social media have affected our social lives and our dietary and lifestyle behaviors [26,27]. The Internet provides immediate access to an enormous amount of information; its use increased and was the main source of information used during the COVID-19 pandemic [28,29]. The Internet has also become the main source of health-related information including nutrition [30]. A valuable aspect of web-based information transactions is the record of communication itself, which, in aggregate, may reflect population-level behaviors. For example, researchers have used search engine queries and volumes to recognize population behavior–based patterns [31]. The data that are accumulated during Internet search activities are one form of Big Data that may provide valuable insights and information into population behavior and interests. One tool that allows users to interact with Internet search data is Google Trends, a free, publicly accessible online portal of Google Inc. Google Trends analyses a portion of the three billion daily Google searches [32]. Google Trends is the most popular tool to gather information on web-based behaviors, and it can be used to predict or prevent health-related issues [33]. The analysis of Internet search queries offers information on the extent of public attention, thereby reflecting the level of public awareness. Google Trends has been used in many research publications to analyze users’ interests across various fields [34].

Currently, limited data are available on how the COVID-19 pandemic affected our dietary and lifestyle-related behaviors and interest in the sustainability of the food system at regional and global levels. Google Trends is a powerful tool to analyze relevant keywords related to these topics and to address this shortcoming. The aim of this study was to investigate the public interest in Food Sustainability and healthy, sustainable dietary patterns across Europe and in European regions using Big Data gathered from Google Trends. More specifically, our objectives were to investigate the changes of public interest in Food Sustainability and healthy sustainable eating patterns between the pre-COVID-19 and post-COVID-19 eras, and to explore the changes in public interest induced by the COVID-19 outbreak by comparing the observed and predicted interest post the COVID-19 outbreak.

## 2. Materials and Methods

### 2.1. Google Trends

Google Trends (https://trends.google.com/trends/ accessed on 11 July 2022) is an online and free tool provided by Google that measures search interest in a particular topic or a set of search terms. It is anonymized, categorized (determining the topic for a search query), and aggregated (grouped into topics). Google Trends data can be accessed in non-real time and go as far back as 2004 and up to 72 h before you search. The data are indexed and normalized, which means the numbers are scaled on a range of 0 to 100, where each data point on the graph represents the proportion of a search topic, as a search term, of total searches in a given country or worldwide during the time period selected. A value of Relative Search Volume (RSV) of 100 is the maximum search interest for the period and location selected. Searches with low volume, duplicate searches, and special characters are excluded from search.

Google trends may qualify analyzed phrases as “search term” or “topic”. Search terms are literally typed words, whereas topics may be proposed by Google Trends when the tool recognizes phrases related to popular queries. Topics enable easy comparison of the given term between countries. For example, the search term “London” will be analyzed by Google Trends exactly; thus, RSV will be the highest in English-speaking countries, whereas the topic “London” will include all queries associated with the query in all the available languages, for example, “Capital of the United Kingdom” and “Londres”, which is “London” in Portuguese and Spanish [30].

It is important to note that none of the searches in the Google database for this study can be associated with a particular individual. The database does not retain information about the identity, IP address, or specific physical location of any user.

Google accounts for >80% of global search engine use [35], and Google Trends is one of the few open sources of search query data. Google Trends is considered a valid and robust indicator to track interest, attention, and public opinion over time [36,37] and has increasingly been used to quantify trends in public interest in several fields such as health care [32,33], tourism [38], and economics [39,40]. The strengths and advantages of Google trends are evident; nevertheless, this tool also presents limitations. Due to time and geographical normalization, we can only compare relative popularity, meaning that different regions that show the same RSV for a term will not always have the same total search volumes. Another limitation to the use of Google Trends is the fact that not everyone uses the Internet, and Internet use may be higher in younger people; therefore, Google Trends data may not be fully representative and may present a selection bias [41,42].

### 2.2. Data Collection

Google Trends searches and data extraction were performed on 11 July 2022, with a single data extraction for each country and covering the period of 1 January 2010 to 30 June 2022. The search period covered before and after the outbreak of the COVID-19 pandemic to reflect changes in relative interests. We used two different approaches: (1) one consisting of the Sustainability, Healthy Diet, “Flexitarianism” (FlexDiet) and “Mediterranean Diet” (MedDiet) topics, and (2) another being a combination of search terms related with “Food Sustainability” (“Food Sustainability” + “sustainable diet” + “sustainable food”). The search term combinations are listed in Table 1 in the official language of each country. The RSVs for the countries of northern Europe (Denmark, Norway, and Sweden), western Europe (France, Germany, Ireland, the Netherlands, and the United Kingdom) and southern Europe (Italy, Portugal, Spain, and Turkey) were retrieved. No RSVs could be retrieved for other European countries for “Food Sustainability”. The filters “Country”, in “All Categories” and for “Google Web Search” were selected in our searches.

### 2.3. Statistical Analysis

The RSVs for each country were used to obtain the monthly mean RSVs for the European region in which the countries were included and the mean RSVs for Europe.

All statistical analyses in the present study were conducted using IBM^®^ SPSS^®^ Statistics for Windows, Version 28.0.1.0 (IBM Corp., Armonk, NY, USA). Forecast models of trend search terms and topics were created using the “Expert Modeler” function. The Autoregressive Integrated Moving Average (ARIMA) and exponential smoothing models were considered to obtain the best-fitting model. Forecast period was set for 30 months, from 1 January 2020 to 30 June 2022, based on the data obtained for the period from 1 January 2010 to 31 December 2019. The best-fitting model for each of the selected dependent variables was selected. Forecasting in the time series for Food Sustainability query, in European regions and in Europe, was performed using Winter’s additive exponential smoothing models. Forecasting in the time series for sustainability topic in southern Europe, northern Europe, and Europe was performed using Winter’s multiplicative exponential smoothing models, and for western Europe was performed using Winter’s additive exponential smoothing model. Regarding the Healthy Diet topic, forecasting in the time series for southern Europe and Europe was performed using Winter’s additive exponential smoothing models, in the time series for western Europe was performed using Winter’s multiplicative exponential smoothing models, and in the time series for northern Europe was performed using simple seasonal exponential smoothing models. Forecasting in the time series for FlexDiet in southern Europe, western Europe, and Europe was performed using Winter’s additive exponential smoothing models, and in the time series for southern Europe was performed using ARIMA (0, 1, 1) (0, 0, 1) model. Forecasting in all the time series for MedDiet was performed using simple seasonal exponential smoothing models.

Differences in RSVs observed 30 months pre- and 30 months post-COVID-19 outbreak were tested (July 2017 to December 2019 vs. January 2020 to June 2022), using the Mann–Whitney U test. The forecast data were then compared to the actual search interest for the same period (January 2020 to June 2022), using the Wilcoxon signed-rank test, to determine the effect of COVID-19 on the selected search term interest. Descriptive statistics are presented as median (*Mdn*) and interquartile range (*IQR*) for each variable. Results were considered statistically significant if *p*-value < 0.05.

## 3. Results

Google Trends data are presented for European regions by term or topic. The RSV evolution for the Food Sustainability query, and the Sustainability, Healthy Diet, FlexDiet, and MedDiet topics, is shown in Figure 1, Figure 2, Figure 3, Figure 4 and Figure 5, respectively.

### 3.1. Trends of RSVs for Food Sustainability

Figure 1 shows the evolution of the RSVs over time for the Food Sustainability query in Europe and European regions. In northern Europe, a significant difference between pre- and post-COVID-19 RSVs for the Food Sustainability query was observed (observed pre-COVID-19: *Mdn* = 29.00, *IQR* = 17.25; observed post-COVID-19: *Mdn* = 38.67, *IQR* = 26.42; *p* = 0.025). No significant differences were identified between post-COVID-19 observed and predicted values (predicted post-COVID-19: *Mdn* = 39.10, *IQR* = 6.59; *p* = 0.543) (Figure 1a).

Higher RSVs for the Food Sustainability query were observed post-COVID-19 compared with pre-COVID-19 in western Europe (observed pre-COVID-19: *Mdn* = 32.30, *IQR* = 12.35; observed post-COVID-19: *Mdn* = 48.10, *IQR* = 19.05; *p* < 0.001). Interestingly, the observed post-COVID-19 RSVs were higher than the predicted post-COVID-19 RSV values (predicted post-COVID-19: *Mdn* = 41.04, *IQR* = 5.81; *p* < 0.001) (Figure 1b).

Higher RSVs for the Food Sustainability query were observed post-COVID-19 compared with pre-COVID-19 in southern Europe (observed pre-COVID-19: *Mdn* = 17.00, *IQR* = 14.69; observed post-COVID-19: *Mdn* = 36.12, *IQR* = 19.69; *p* < 0.001). Observed post-COVID-19 RSVs were higher than predicted post-COVID-19 RSV values (predicted post-COVID-19: *Mdn* = 26.29, *IQR* = 5.43; *p* < 0.001) (Figure 1c).

Considering the mean values of the 12 European countries, higher values were observed post-COVID-19 compared with pre-COVID-19 (observed pre-COVID-19: *Mdn* = 27.29, *IQR* = 10.90; observed post-COVID-19: *Mdn* = 42.96, *IQR* = 18.71; *p* < 0.001). Despite the absence of a significant difference in the northern European region, lower post-COVID-19 RSV values were predicted when compared with the observed values (predicted post-COVID-19: *Mdn* = 35.90, *IQR* = 5.17; *p* < 0.001) (Figure 1d).

### 3.2. Trends of RSVs for Sustainability

Figure 2 shows the evolution of RSVs for the Sustainability topic. In northern Europe, higher post-COVID-19 RSVs for the Sustainability topic were observed compared with pre- COVID-19 RSVs (observed pre-COVID-19: *Mdn* = 53.67, *IQR* = 17.75; observed post-COVID-19: *Mdn* = 76.50, *IQR* = 21.00; *p* < 0.001). Despite the higher observed post-COVID-19 RSVs, those values were lower than what was predicted by our model (predicted post-COVID-19: *Mdn* = 88.12, *IQR* = 27.21; *p* < 0.001) (Figure 2a).

In western Europe, higher RSVs for the Sustainability topic were observed post-COVID-19 compared with pre-COVID-19 (observed pre-COVID-19: *Mdn* = 37.60, *IQR* = 12.95; observed post-COVID-19: *Mdn* = 58.40, *IQR* = 19.30; *p* < 0.001). Once again, observed post-COVID-19 RSVs were lower than predicted post-COVID-19 RSV values (predicted post-COVID-19: *Mdn* = 64.35, *IQR* = 16.70; *p* < 0.001) (Figure 2b).

Higher RSVs for the Sustainability topic were observed post-COVID-19 compared with pre-COVID-19 in southern Europe (observed pre-COVID-19: *Mdn* = 43.75, *IQR* = 15.00; observed post-COVID-19: *Mdn* = 68.50, *IQR* = 22.75; *p* < 0.001). Similar to what was observed for northern and western European regions, in southern Europe, observed post-COVID-19 RSVs were lower than predicted post-COVID-19 RSV values (predicted post-COVID-19: *Mdn* = 77.39, *IQR* = 21.89; *p* < 0.001) (Figure 2c).

Along the same lines as the above-mentioned results, higher values were observed post-COVID-19 compared with pre-COVID-19 considering the mean values of the 12 European countries (observed pre-COVID-19: *Mdn* = 44.83, *IQR* = 15.92; observed post-COVID-19: *Mdn* = 66.25, *IQR* = 17.94; *p* < 0.001). Also, higher post-COVID-19 RSV values were predicted when compared with the observed values (predicted post-COVID-19: *Mdn* = 72.30, *IQR* = 17.63; *p* < 0.001) (Figure 2d).

### 3.3. Trends of RSVs for Healthy Diet

The evolution of the RSVs for the Healthy Diet topic is shown in Figure 3. In northern Europe, no significant differences in RSVs pre- and post-COVID-19 for the Healthy Diet topic were observed (observed pre-COVID-19: *Mdn* = 60.50, *IQR* = 9.75; observed post-COVID-19: *Mdn* = 59.33, *IQR* = 21.00; *p* = 0.456). The predicted post-COVID-19 RSVs for the Healthy Diet topic were modestly, yet significantly, higher than the observed values (predicted post-COVID-19: *Mdn* = 61.79, *IQR* = 9.21; *p* = 0.022) (Figure 3a).

In western Europe, no significant differences in RSVs pre- and post-COVID-19 for the Healthy Diet topic were observed (observed pre-COVID-19: *Mdn* = 54.90, *IQR* = 6.95; observed post-COVID-19: *Mdn* = 52.20, *IQR* = 15.10; *p* = 0.408). Also, the observed post-COVID-19 RSVs were not significantly different from the predicted post-COVID-19 RSV values (predicted post-COVID-19: *Mdn* = 54.90, *IQR* = 11.08; *p* = 0.088) (Figure 3b).

Lower RSVs for the Healthy Diet topic were observed post-COVID-19 compared with pre-COVID-19 in southern Europe (observed pre-COVID-19: *Mdn* = 60.00, *IQR* = 13.13; observed post-COVID-19: *Mdn* = 50.37, *IQR* = 11.81; *p* < 0.001). The observed post-COVID-19 RSVs were lower than the predicted post-COVID-19 RSV values (predicted post-COVID-19: *Mdn* = 63.80, *IQR* = 15.18; *p* < 0.001) (Figure 3c).

Considering the mean values of the 12 European countries, no differences were observed post-COVID-19 compared with pre-COVID-19 RSVs for the Healthy Diet topic (observed pre-COVID-19: *Mdn* = 57.29, *IQR* = 6.71; observed post-COVID-19: *Mdn* = 53.67, *IQR* = 13.23; *p* = 0.055). Higher post-COVID-19 RSV values were predicted when compared with the observed values (predicted post-COVID-19: *Mdn* = 62.38, *IQR* = 11.29; *p* < 0.001) (Figure 3d).

### 3.4. Trends of RSVs for FlexDiet

Figure 4 presents the evolution of RSVs for the FlexDiet topic in Europe and European regions. In northern Europe, lower post-COVID-19 RSVs for the FlexDiet topic were observed compared with pre-COVID-19 RSVs (observed pre-COVID-19: *Mdn* = 31.33, *IQR* = 12.08; observed post-COVID-19: *Mdn* = 19.83, *IQR* = 16.50; *p* < 0.001). The observed post-COVID-19 RSVs were also lower than what was predicted by our model (predicted post-COVID-19: *Mdn* = 36.02, *IQR* = 8.35; *p* < 0.001) (Figure 4a).

In western Europe, no significant differences in RSVs pre- and post-COVID-19 for the FlexDiet topic were observed (observed pre-COVID-19: *Mdn* = 26.30, IQR = 20.85; observed post-COVID-19: *Mdn* = 33.60, *IQR* = 8.80; *p* = 0.068). The observed post-COVID-19 RSVs were lower than the predicted post-COVID-19 RSV values (predicted post-COVID-19: *Mdn* = 45.40, *IQR* = 4.28; *p* < 0.001) (Figure 4b).

Higher RSVs for the FlexDiet topic were observed post-COVID-19 compared with pre-COVID-19 in southern Europe (observed pre-COVID-19: *Mdn* = 22.12, *IQR* = 17.69; observed post-COVID-19: *Mdn* = 27.62, *IQR* = 14.56; *p* < 0.001). Despite the increase in observed post-COVID-19 RSVs, those values were lower than what was predicted (predicted post-COVID-19: *Mdn* = 32.34, *IQR* = 1.80; *p* < 0.001) (Figure 4c).

The mean values of the 12 European countries show no differences when comparing observed post-COVID-19 and pre-COVID-19 RSVs for the FlexDiet topic (observed pre-COVID-19: *Mdn* = 28.25, *IQR* = 12.52; observed post-COVID-19: *Mdn* = 29.00, *IQR* = 9.04; *p* = 0.502). Consistent with what was observed in all the European regions, higher post-COVID-19 RSV values were predicted when compared with the observed values (predicted post-COVID-19: *Mdn* = 39.97, *IQR* = 4.35; *p* < 0.001) (Figure 4d).

### 3.5. Trends of RSVs for MedDiet

The evolution of the RSVs for the MedDiet topic is shown in Figure 5. In northern Europe, no significant differences between pre- and post-COVID-19 RSVs for the MedDiet topic were observed (observed pre-COVID-19: *Mdn* = 22.67, *IQR* = 11.67; observed post-COVID-19: *Mdn* = 26.00, *IQR* = 9.33; *p* = 0.084). The observed and predicted post-COVID-19 RSVs were also not significantly different (predicted post-COVID-19: *Mdn* = 25.262, *IQR* = 4.17; *p* = 0.309) (Figure 5a).

In western Europe, the observed post-COVID-19 RSVs for the MedDiet topic were higher than the pre-COVID-19 RSVs (observed pre-COVID-19: *Mdn* = 29.90, *IQR* = 11.55; observed post-COVID-19: *Mdn* = 40.20, *IQR* = 13.25; *p* < 0.001). The observed post-COVID-19 RSVs were higher than the predicted post-COVID-19 RSV values (predicted post-COVID-19: *Mdn* = 32.96, *IQR* = 4.10; *p* < 0.001) (Figure 5b).

Higher RSVs for the MedDiet topic were observed post-COVID-19 compared with pre-COVID-19 in southern Europe (observed pre-COVID-19: *Mdn* = 30.62, *IQR* = 10.75; observed post-COVID-19: *Mdn* = 34.75, *IQR* = 13.19; *p* = 0.024). No significant differences in observed and predicted post-COVID-19 RSVs were identified (predicted post-COVID-19: *Mdn* = 35.67, *IQR* = 8.97; *p* = 0.824) (Figure 5c).

Considering the mean values of the 12 European countries, higher observed post-COVID-19 RSVs were found compared to pre-COVID-19 RSVs for the MedDiet topic (observed pre-COVID-19: *Mdn* = 28.21, *IQR* = 7.73; observed post-COVID-19: *Mdn* = 35.00, *IQR* = 7.96; *p* < 0.001). The observed post-COVID-19 values were also higher than the predicted post-COVID-19 RSV values (predicted post-COVID-19: *Mdn* = 31.13, *IQR* = 4.98; *p* < 0.001) (Figure 5d).

## 4. Discussion

In this study, we explored, for the first time, the impact of the COVID-19 pandemic on the public’s interest in Food Sustainability, in Sustainability, and in dietary patterns, namely the Healthy Diet, FlexDiet, and MedDiet, using Big Data. Specifically, we used the RSVs provided by Google Trends to obtain temporal series of public interest in the mentioned topics before the COVID-19 pandemic and to forecast public interest after it. By comparing the predicted and observed values, an image of how this global health event affected public interest in Food Sustainability, in Sustainability, and in dietary patterns, was obtained. Using this strategy, an increase in Food Sustainability public interest after the COVID-19 outbreak was observed, and this increase was higher than what was forecast based on the previous data. A significant increase in interest in Sustainability in general was also observed; nevertheless, this increase was smaller than that forecast by our models. Taken together, these results may indicate that the COVID-19 pandemic outbreak sparked an interest in Food Sustainability while reducing the increase in interest in general Sustainability. Furthermore, mixed results regarding dietary patterns were obtained across European regions; yet, considering the mean interest for Europe, it seems that the COVID-19 pandemic outbreak dampened the interest in dietary patterns such as the Healthy Diet and FlexDiet, and promoted interest in the MedDiet, at least in the western European region.

### 4.1. Changes in Food Sustainability Interest after COVID-19

The COVID-19 pandemic has changed the functioning of food systems all over the world and, despite its stability, Europe has been no exception. The pandemic outbreak caused problems in the operations of all the food system’s players, including the actors providing the means of food production, the food producers, the logistics and processing industry and, inevitably, this turbulence went all the way to consumers. The COVID-19 pandemic affected the food system directly and indirectly; for instance, COVID-19 disrupted food supply transportation (e.g., access to container transport) and changed the demands for related services (such as retail food pickup and delivery services) [43]. Incidents of food flow disruption due to the pandemic were largely highlighted by the media and varied from vegetables rotting in fields and milk being dumped, to food processing facilities running short of workers due to the disease, and panic buying in stores. Strains on multiple points of the food supply chain could affect food availability and prices in the retail sector, ultimately impacting people at the receiving end [44,45]. Our results indicate that public interest in Food Sustainability increased after the COVID-19 outbreak to a higher degree than would be expected considering the tendencies of previous years. A possible explanation may reside in the perfect storm that assaulted the food system during the pandemic and the media coverage of those problems that may have led consumers to seek information on food sustainability and security.

Few studies have explored the adoption of sustainable food behaviors and attitudes during the COVID-19 pandemic. Muresan et al. [46] explored consumers’ attitudes towards sustainable food consumption during the COVID-19 pandemic in Romania and found a positive attitude towards sustainable food behavior. Despite the positive attitude towards sustainable food behavior, as observed in previous works [47,48], a gap between attitudes and actual behaviors was observed. The authors also noted that respondents supported local producers, were aware of the importance of domestic agriculture, and were concerned about their health and waste reduction when planning a food menu; however, they also noted that even if the respondents’ perception of the sustainable food choices was positive, their purchasing behavior was very different when it came to buying local products or supporting third-world farmers. Possible explanations were put forward, such as the higher prices of local foods compared to prices in supermarket chains or simply the fact that Romanian consumers are not ready to change their consumption patterns, as part of a complex and long process (51). For the most part, human behavior is the force of habits determined by cultural, social, and economic factors. Habits are forged in the process of repeating the same responses in particular contexts. Dietary choices are made virtually every day, driven by habitual response, and are extremely stable [49]. However, when the components that make up our environment or circumstances change, our food choice habits may change as well [50]. Therefore, it is possible that the pandemic period, which patently constitutes a disruption to habitual circumstances, could provoke dietary habit changes [49].

### 4.2. COVID-19 Impact on Sustainability Interest

Muresan et al. [46] also reported that respondents remained neutral towards the environmental issue, without any interest in participating in protection actions, with their actual behavior being limited to recycling food packaging. Along the same lines, Rousseau and Deschacht [51], by analyzing online search behavior in twenty European countries regarding how public awareness of nature and the environment has evolved during the COVID-19 crisis, found that public awareness of environmental issues such as climate change was unaffected by the COVID-19 crisis based on online searches. Our results are along the same lines; despite significantly higher levels of interest in Sustainability after COVID-19, the increase was smaller than would be expected if the pre-COVID-19 conditions were maintained. A possible explanation may reside in the perception that the general increasing sense of urgency observed in the last few years for many environmental problems may have been counteracted by the reduction in media attention and the short-term positive trends reported for several pollution levels [51].

### 4.3. COVID-19 Impact on Dietary Pattern Interest

Surprisingly, public interest in the Healthy Diet and FlexDiet decreased after the COVID-19 pandemic outbreak. These results are unexpected, since health is one of the main concerns that consumers have in mind when making food choices, especially during COVID-19 times [52]. No studies were found regarding the adherence to healthy or flexitarian dietary patterns, yet one study reported an increase in meat consumption after the pandemic outbreak [53]. Furthermore, reports from gray literature indicate a decline in the number of flexitarians in the United Kingdom [54]. We also observed an increase in the public interest in the MedDiet that was greater than would be expected. In agreement with our results, Della Valle and collaborators [55] found evidence suggesting that adherence to the MedDiet during lockdown might have increased in some settings, while the determinants of such a trend are to be further explored.

No data are available regarding the media coverage of dietary patterns before and/or after the COVID-19 outbreak. However, it is possible that the large media attention directed to COVID-19 may have diverted public attention away from dietary-related issues. It is also likely that media coverage on dietary issues varies across European regions and that, along with many other factors, may explain the inconsistent results obtained across European regions.

### 4.4. Strengths and Limitations

Despite the wide use of Google trends to assess public interest in several fields, no research was found regarding Food Sustainability and/or sustainable diets. The assessment of the sustainability of diets is a topic that has received a sizable amount of research attention in recent years. Most of the research conducted in that field has been performed since 2006 [56]. Consumers are relevant players, shaping food systems [22,57,58], and while the academic interest in this topic is clear, public/consumer interest seems to be overlooked [59].

The novelty of this work is a strength, yet some limitations must be acknowledged. Google Trends is considered a valid and robust indicator to track interest, attention, and public opinion over time [36,37]. Nevertheless, Google Trends is not transparent regarding what content is included in each topic. It is possible that the content included in a topic, despite being related, may not be the most obvious. For instance, it is reasonable to assume that searches related to the history or the economy of the MedDiet are included in the MedDiet topic. Despite being related, those searches are not relevant to the MedDiet as a dietary pattern. Furthermore, while the topics of MedDiet and Flexitarianism cover well-defined and recognizable diets, the topic of Healthy Diet may include a broad range of diets, including MedDiet and Flexitarianism. A strict interpretation of the topics used in this study as dietary patterns should be made with caution and considering they are a proxy measure.

## 5. Conclusions

This work explored, for the first time, changes in consumer interest in Food Sustainability, Sustainability, and healthy dietary patterns in Europe induced by COVID-19. Across the counties analyzed, public interest in Food Sustainability increased as a result of the COVID-19 outbreak, and this increase was not the result of a general increase in public interest in Sustainability. Also, public interest in healthy dietary patterns was reduced or only regionally increased.

Understanding consumers’ behavior and interest toward food choices is crucial for the transition to sustainable diets, and to the definition of the educational and behavioral interventions that are essential to this transition [60]. Engagement, interest, and information seeking are also essential for the construction of beliefs and behaviors. More research is needed in this field; nevertheless, these findings may be important for policy makers to design and select strategies that may promote a transition to a sustainable food system supported by healthy dietary patterns.

## Figures and Tables

**Figure 1 nutrients-14-04898-f001:**
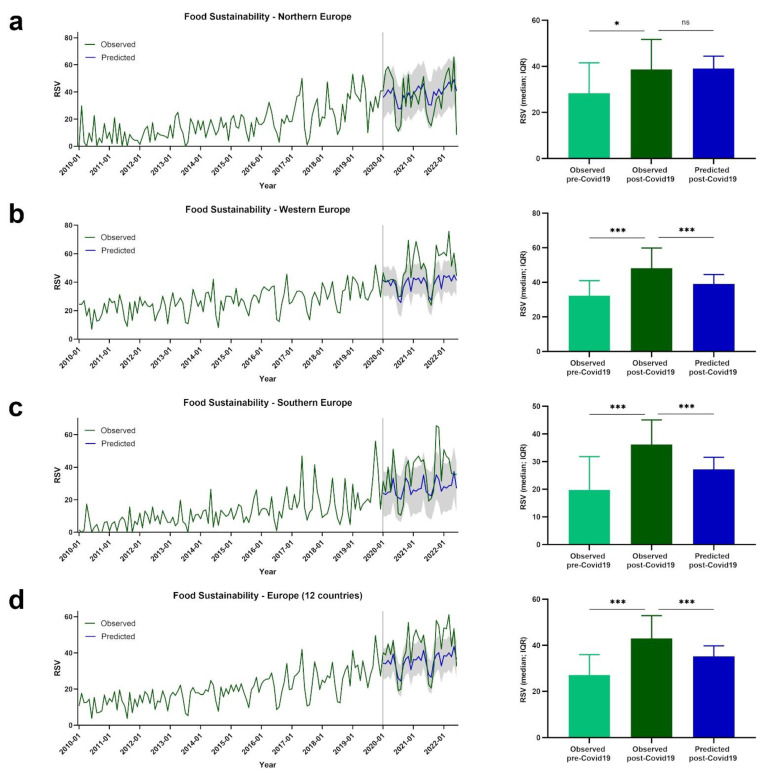
The RSV evolution for Food Sustainability query. (**a**) Evolution of RSVs for Food Sustainability query in northern Europe from January 2010 to June 2022 (green line), predicted evolution of RSVs for Food Sustainability query between January 2020 and June 2022 (blue line); gray shadow represents 95% confidence interval. (**b**) Evolution of RSVs for Food Sustainability query in western Europe from January 2010 to June 2022 (green line), predicted evolution of RSVs for Food Sustainability query between January 2020 and June 2022 (blue line); gray shadow represents 95% confidence interval. (**c**) Evolution of RSVs for Food Sustainability query in southern Europe from January 2010 to June 2022 (green line), predicted evolution of RSVs for Food Sustainability query between January 2020 and June 2022 (blue line); gray shadow represents 95% confidence interval. (**d**) Evolution of RSVs for Food Sustainability query in Europe from January 2010 to June 2022 (green line), predicted evolution of RSVs for Food Sustainability query between January 2020 and June 2022 (blue line); gray shadow represents 95% confidence interval. For (**a**–**d**) bars represent the comparison between median RSVs pre-COVID-19 and post-COVID-19, and between post-COVID-19 and predicted post-COVID-19 data. * *p* < 0.05; *** *p* < 0.001; ns—not significant.

**Figure 2 nutrients-14-04898-f002:**
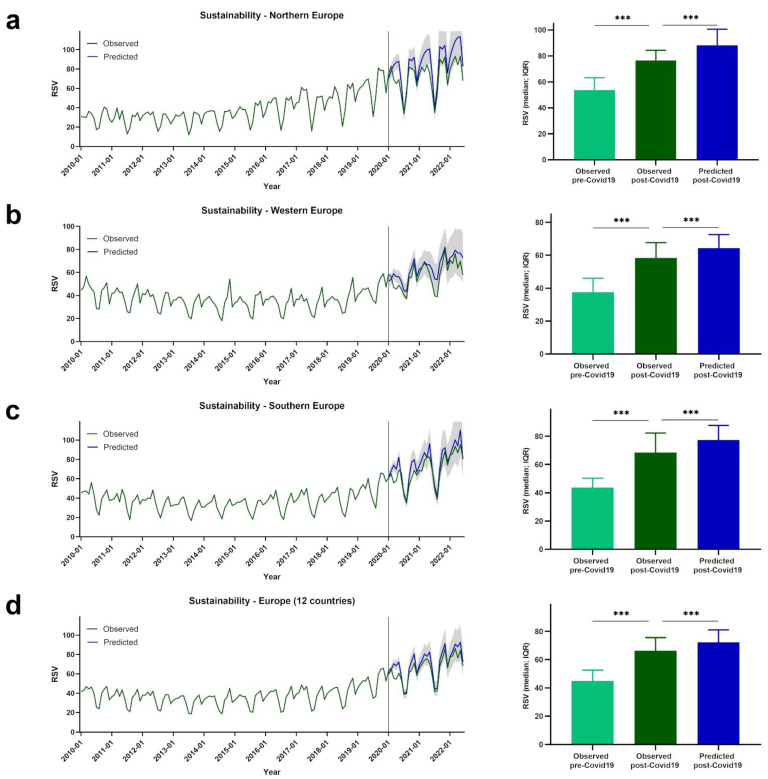
The RSV evolution for Sustainability topic. (**a**) Evolution of RSVs for Sustainability topic in northern Europe from January 2010 to June 2022 (green line), predicted evolution of RSVs for Sustainability topic between January 2020 and June 2022 (blue line); gray shadow represents 95% confidence interval. (**b**) Evolution of RSVs for Sustainability topic in western Europe from January 2010 to June 2022 (green line), predicted evolution of RSVs for Sustainability topic between January 2020 and June 2022 (blue line); gray shadow represents 95% confidence interval. (**c**) Evolution of RSVs for Sustainability topic in southern Europe from January 2010 to June 2022 (green line), predicted evolution of RSVs for Sustainability topic between January 2020 and June 2022 (blue line); gray shadow represents 95% confidence interval. (**d**) Evolution of RSVs for Sustainability topic in Europe from January 2010 to June 2022 (green line), predicted evolution of RSVs for Sustainability topic between January 2020 and June 2022 (blue line); gray shadow represents 95% confidence interval. For (**a**–**d**) bars represent the comparison between median RSVs pre-COVID-19 and post-COVID-19, and between post-COVID-19 and predicted post-COVID-19 data. *** *p* < 0.001.

**Figure 3 nutrients-14-04898-f003:**
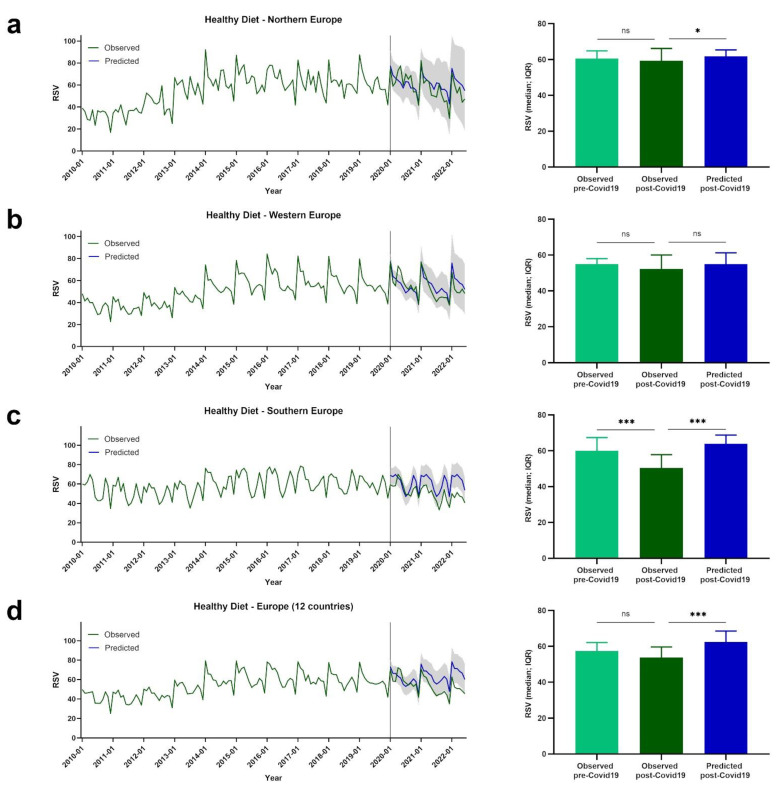
The RSV evolution for Healthy Diet topic. (**a**) Evolution of RSVs for Healthy Diet topic in northern Europe from January 2010 to June 2022 (green line), predicted evolution of RSVs for Healthy diet topic between January 2020 and June 2022 (blue line); gray shadow represents 95% confidence interval. (**b**) Evolution of RSVs for Healthy Diet topic in western Europe from January 2010 to June 2022 (green line), predicted evolution of RSVs for Healthy Diet topic between January 2020 and June 2022 (blue line); gray shadow represents 95% confidence interval. (**c**) Evolution of RSVs for Healthy Diet topic in southern Europe from January 2010 to June 2022 (green line), predicted evolution of RSVs for Healthy Diet topic between January 2020 and June 2022 (blue line); gray shadow represents 95% confidence interval. (**d**) Evolution of RSVs for Healthy Diet topic in Europe from January 2010 to June 2022 (green line), predicted evolution of RSVs for Healthy Diet topic between January 2020 and June 2022 (blue line); gray shadow represents 95% confidence interval. For (**a**–**d**) bars represent the comparison between median RSVs pre-COVID-19 and post-COVID-19, and between post-COVID-19 and predicted post-COVID-19 data. * *p* < 0.05; *** *p* < 0.001; ns—not significant.

**Figure 4 nutrients-14-04898-f004:**
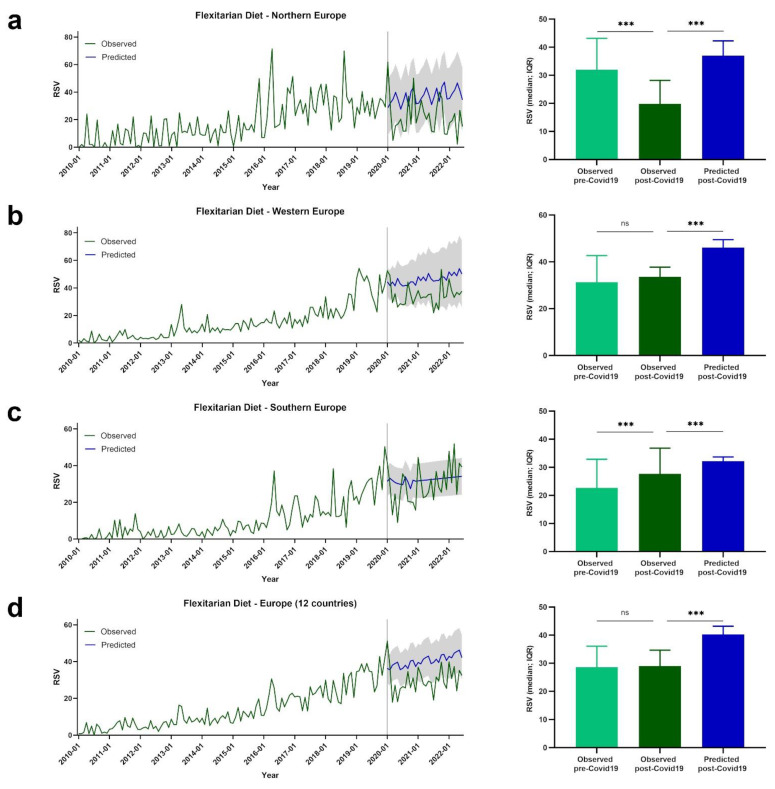
The RSV evolution for FlexDiet topic. (**a**) Evolution of RSVs for FlexDiet topic in northern Europe from January 2010 to June 2022 (green line), predicted evolution of RSVs for FlexDiet topic between January 2020 and June 2022 (blue line); gray shadow represents 95% confidence interval. (**b**) Evolution of RSVs for FlexDiet topic in western Europe from January 2010 to June 2022 (green line), predicted evolution of RSVs for FlexDiet topic between January 2020 and June 2022 (blue line); gray shadow represents 95% confidence interval. (**c**) Evolution of RSVs for FlexDiet topic in southern Europe from January 2010 to June 2022 (green line), predicted evolution of RSVs for FlexDiet topic between January 2020 and June 2022 (blue line); gray shadow represents 95% confidence interval. (**d**) Evolution of RSVs for FlexDiet topic in Europe from January 2010 to June 2022 (green line), predicted evolution of RSVs for FlexDiet topic between January 2020 and June 2022 (blue line); gray shadow represents 95% confidence interval. For (**a**–**d**) bars represent the comparison between median RSVs pre-COVID-19 and post-COVID-19, and between post-COVID-19 and predicted post-COVID-19 data. *** *p* <.001; ns—not significant.

**Figure 5 nutrients-14-04898-f005:**
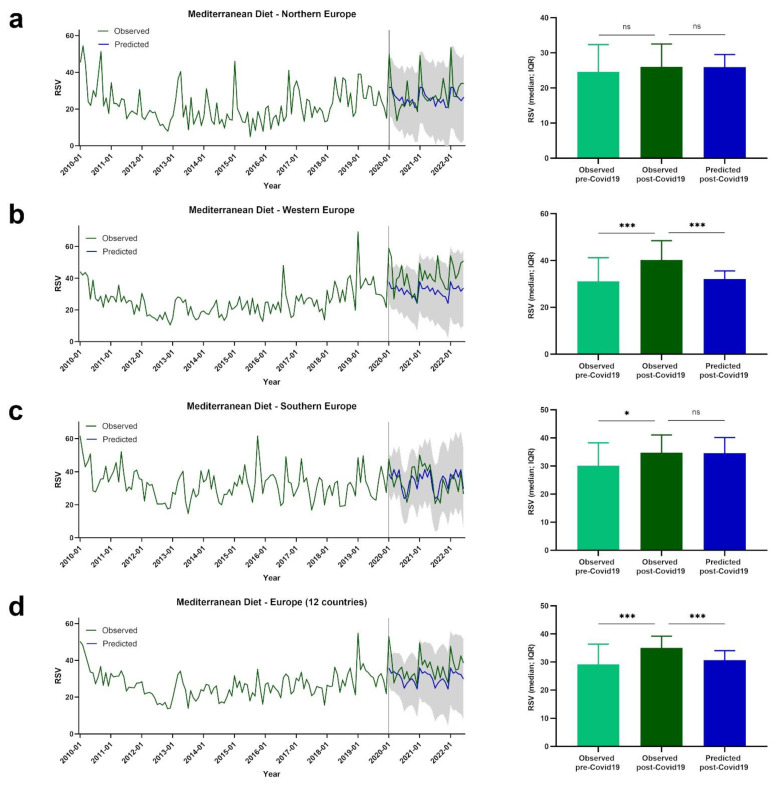
The RSV evolution for MedDiet topic. (**a**) Evolution of RSVs for MedDiet topic in northern Europe from January 2010 to June 2022 (green line), predicted evolution of RSVs for MedDiet topic between January 2020 and June 2022 (blue line); gray shadow represents 95% confidence interval. (**b**) Evolution of RSVs for MedDiet topic in western Europe from January 2010 to June 2022 (green line), predicted evolution of RSVs for MedDiet topic between January 2020 and June 2022 (blue line); gray shadow represents 95% confidence interval. (**c**) Evolution of RSVs for MedDiet topic in southern Europe from January 2010 to June 2022 (green line), predicted evolution of RSVs for MedDiet topic between January 2020 and June 2022 (blue line); gray shadow represents 95% confidence interval. (**d**) Evolution of RSVs for MedDiet topic in Europe from January 2010 to June 2022 (green line), predicted evolution of RSVs for MedDiet topic between January 2020 and June 2022 (blue line); gray shadow represents 95% confidence interval. For (**a**–**d**) bars represent the comparison between median RSVs pre-COVID-19 and post-COVID-19, and between post-COVID-19 and predicted post-COVID-19 data. * *p* < 0.05; *** *p* < 0.001; ns—not significant.

**Table 1 nutrients-14-04898-t001:** The search term combinations for “Food Sustainability”.

Country	Language	Search Term Queries
Denmark	Danish	Mad Bæredygtighed + Bæredygtig kost + Bæredygtig mad
Norway	Norwegian	Mat bærekraft + bærekraftig kosthold + bærekraftig mat
Sweden	Swedish	Livsmedelshållbarhet + Hållbar kost + Hållbar mat
France	French	Durabilité alimentaire + Alimentation durable
Germany	German	Lebensmittelnachhaltigkeit + Nachhaltige Ernährung + Nachhaltige Lebensmittel
The Netherlands	Dutch	Voedselduurzaamheid + duurzaam dieet + duurzaam voedsel
The United Kingdom	English	Food Sustainability + Sustainable Diet + Sustainable Food
Ireland	English	Food Sustainability + Sustainable Diet + Sustainable Food
Portugal	Portuguese	Sustentabilidade Alimentar + Dieta Sustentável + Alimentação Sustentável
Spain	Spanish	Sostenibilidad alimentaria + Dieta sostenible + Alimentación sostenible
Italy	Italian	Sostenibilità Alimentare + Dieta Sostenibile + Alimentazione Sostenibile
Turkey	Turkish	Gıda Sürdürülebilirliği + Sürdürülebilir Diyet + Sürdürülebilir Gıda

## Data Availability

Not applicable.

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
