# Peer review of "COVID-19 Changes Public Awareness about Food Sustainability and Dietary Patterns: A Google Trends Analysis"

_nutrients, 2022, doi:10.3390/nu14224898_

Round 1

Reviewer 1 Report

This is an interesting paper, however, there are some areas that need improvement or greater clarification. Please see comments below divided by section:

Abstract: 

1. I'm a bit confused about the timing of the data used - why is 2010 mentioned? weren't the data from 2017-2022?

Introduction: 

1. I would advise editing for English,

2. Also, I would recommend restructuring/streamlining the introduction for the flow of the paper. The authors have clearly developed a well-researched manuscript, however, it is almost too detailed and seems to stray from the main purpose of the paper. 

 Methods: 

1. Why was the 30-month time period selected?

2. Were the various assumptions needed to run these analyses explored? Please explain. 

Results: 

1. Figures need to be larger to be more legible to the reader. 

2. Consider streamlining the detail of the captions and discussing that more in the narrative text. 

Discussion: 

1. Consider the use of subheadings for greater flow and reader ease. 

Again, this is a really interesting paper. However, I think these edits will help streamline the paper and better present the interesting results. 

Author Response

Comments and Suggestions for Authors
This is an interesting paper, however, there are some areas that need improvement or
greater clarification. Please see comments below divided by section:

We appreciate the comment to our work, and we would like to thank the very interesting
and pertinent suggestion that the reviewer made. We believe that they significantly
contribute to the improvement of the work.

Abstract:

1. I'm a bit confused about the timing of the data used - why is 2010 mentioned? weren't
the data from 2017-2022?

In the construction the forecasting models (time series models) enough time points must
be used to ensure that patterns are correctly identified and reproduced. Our forecasting
models rely on data from January 2010 to December 2019 (120 months) to predict the
data for the period comprised January 2020 to June 2022 (30 months). Briefly, data was
obtained from January 2010 to June 2022, then we considered data from January 2010 to
December 2019 as pre-Covid-19 data and used those data to forecast the data behaviour
for the post-Covid-19 period (set from January 2020 to June 2022). We compared the 30
months post-Covid -19 with the last 30 months of the pre-Covid-19 period (July 2017 to
December 2019). The period between July 2017 to December 2019 was chose to have the
same duration as the period that was being compared. We also tested the real data from
the post-Covid-19 period with the post-Covid-19 forecasted data.

We understand that our strategy is not clearly presented in the abstract due to space
constrains.

Introduction:

1. I would advise editing for English,

The paper was fully revised for English grammar and punctuation.

2. Also, I would recommend restructuring/streamlining the introduction for the flow of
the paper. The authors have clearly developed a well-researched manuscript; however, it
is almost too detailed and seems to stray from the main purpose of the paper.

We made some changes to the introduction and tried to remove unessential parts. We also
made some alterations to accommodate the suggestions of other reviewer.

Methods:

1. Why was the 30-month time period selected?

We assume that the question is regarding the forecast time period. The 30 months period
was chosen because it was the time between the Covid-19 outbreak and the available data
(January 2020 to June 2022). We forecast the data for that period based on the data from
January 2010 to December 2019 (pre-Covid-19 data) to predict what would have happen
if there was no Covid-19 outbreak. The median value for the data forecasted for this
period was compared with the median value of the 30 months’ period immediately before
(July 2017 to December 2019) and with the real data collected for that period.

2. Were the various assumptions needed to run these analyses explored? Please explain.

Two major assumptions were needed to run these analyses:

Assumptions 1. The dependent variable and any independent variables are treated as time
series, meaning that each case represents a time point, with successive cases separated by
a constant time interval.

Assumptions 2. The variables should not contain any embedded missing data. At least
one periodic date component must be defined.

Those assumptions were met and were ensured by the data extraction made.

Furthermore, the expert modeler capability of the SPSS software will give the best fitted
model for the data that was imputed. Measures of goodness-of-fit were checked
individually for each model (stationary R-square, R-square (R 2), root mean square error
(RMSE), mean absolute error (MAE), mean absolute percentage error (MAPE),
maximum absolute error (MaxAE), maximum absolute percentage error (MaxAPE), and
normalized Bayesian information criterion (BIC).

Results:
1. Figures need to be larger to be more legible to the reader.

We altered the figures according to the suggestion. We believe that now they are much
clear and appealing. Thanks for the suggestion.

2. Consider streamlining the detail of the captions and discussing that more in the
narrative text.

The description of the models used in forecasting was moved from captions to the
statistical section.

Discussion:

1. Consider the use of subheadings for greater flow and reader ease.

We agree with your suggestion and subheadings were added accordingly.

Again, this is a really interesting paper. However, I think these edits will help streamline
the paper and better present the interesting results.

Once again, we appreciate the comment to our work, and we would like to thank the very
interesting and pertinent suggestion that the reviewer made. We believe that they
significantly contribute to the improvement o
f the work.

Reviewer 2 Report

In this manuscript, the authors report an analysis of changes in public interest in food sustainability and dietary patterns across Europe using trends in search engine data from Google.  Clarification of the following points would be helpful.

The specific dietary patterns the authors chose to focus on are “Healthy Diet”, “Mediterranean diet” and “flexitarianism”.  It would be helpful for the authors to provide a bit more background on how these specific diet patterns are related to sustainability. It would also be helpful for the authors to provide very brief description of these particularly dietary patterns.

Healthy Diet is quite vague and general and evaluation of “Healthy Diet” as a dietary pattern may not be appropriate as there is substantial overlap with the other dietary patterns evaluated.   “Mediterranean Diet”, which is a well-established and specific dietary pattern, and in lines 74-75, the authors indicate that a flexitarian diet describes a “healthy reference diet”.  Do the authors consider the “Healthy Diet” pattern to be distinct from the Mediterranean and Flexitarian diets in some way?  If not – then it should not be described as a specific Dietary pattern, and should be discussed/evaluated as a general topic of healthy diet (similar to sustainability).

Page 8, lines 271-274:  Some of this statistical modeling may be more appropriate for the statistical analysis section.

Discussion – paragraph 1.  It seems plausible that some of the mixed results for dietary patterns related to specific European regions might be related to the focus of local media.  The authors discuss this phenomenon in lines 446-449 related to sustainability – but the same could be true for dietary patterns.

Lines 78-79: the authors list several factors that influence consumers food choices and dietary behavior, of which “biological determinants (genes) is listed first.  It might be better to move that to later in the list, as the other factors listed are likely to have a larger influence on dietary behavior than genetics.

Title – Use of the word “awareness” is somewhat misleading as it suggests knowledge; whereas the authors evaluated interest via Google search trends.

Author Response

In this manuscript, the authors report an analysis of changes in public interest in food
sustainability and dietary patterns across Europe using trends in search engine data from
Google. Clarification of the following points would be helpful.

We would like to thank the very interesting and pertinent suggestion that the reviewer
made. We believe that they significantly contribute to the improvement of the work.

The specific dietary patterns the authors chose to focus on are “Healthy Diet”,
“Mediterranean diet” and “flexitarianism”. It would be helpful for the authors to provide
a bit more background on how these specific diet patterns are related to sustainability. It
would also be helpful for the authors to provide very brief description of these particularly
dietary patterns.

Thanks for the suggestion. We added in the introduction some background on those
dietary patterns and a reference to their mitigation potential on greenhouse gas emissions.

A brief description of the dietary patterns was also added. This addition was particularly
relevant since Healthy diet is a broad term that include many dietary patterns that can be
distinct. More on this topic will be addressed in the next commentary.

Healthy Diet is quite vague and general and evaluation of “Healthy Diet” as a dietary
pattern may not be appropriate as there is substantial overlap with the other dietary
patterns evaluated. “Mediterranean Diet”, which is a well-established and specific

dietary pattern, and in lines 74-75, the authors indicate that a flexitarian diet describes a
“healthy reference diet”. Do the authors consider the “Healthy Diet” pattern to be distinct
from the Mediterranean and Flexitarian diets in some way? If not then it should not be
described as a specific Dietary pattern, and should be discussed/evaluated as a general
topic of healthy diet (similar to sustainability).

We agree with the reviewer that healthy diet is a brad term. Our strategy was based on
the dietary patterns described in the IPCC report on climate change and land from 2019
(chapter 5, pag. 488) where the mitigation potential on greenhouse gas emission of
different diets is presented. Based on those dietary patterns we chose the ones that did not
exclude any food group (vegan, vegetarian, pescetarian were no included due to the
exclusion of meat and other foods) and were widely known and used (fair and frugal and
climate carnivore were not included due to limited information could be retrieved from
google trends). On the IPCC report Healthy diet is defined as being based on global
dietary guidelines for consumption of red meat, sugar, fruits and vegetables, and total
energy intake. Independently of the distinctions made on the IPCC report, we
acknowledge that MedDiet and flexitarianism are indeed healthy diets, each one with
their specifical characteristics. Google trends topic may not discriminate the overlap
between those dietary patterns and that is a limitation of our strategy. Also, Google trends
is not transparent regarding what may be included in the topics that are presented and that
limitation should also be mentioned. Tacking in consideration all the limitations regarding
the use of google trends topics we admit that the three dietary patterns should be
interpreted as general topics. A section on strengths and limitations was added and the
above-mentioned points were discussed in that section.

We further added a correction in the introduction since where it was read “healthy
reference diet” it should be read “Planetary healthy reference diet”.

Page 8, lines 271-274: Some of this statistical modelling may be more appropriate for
the statistical analysis section.

We appreciate the suggestion. All the information present in figure captions regarding
statistical modelling was moved to the statistical analysis section.

Discussion paragraph 1. It seems plausible that some of the mixed results for dietary
patterns related to specific European regions might be related to the focus of local media.
The authors discuss this phenomenon in lines 446-449 related to sustainability but the
same could be true for dietary patterns.

That is a very interesting observation. Indeed, we also think that it can be true for food
sustainability, unfortunately we don’t have any way to support that. Nevertheless, we
added that to the discussion, in the section 4.3. Covid-19 impact in dietary patterns
interest as a possibility.

Lines 78-79: the authors list several factors that influence consumers food choices and
dietary behavior, of which “biological determinants (genes) is listed first. It might be
better to move that to later in the list, as the other factors listed are likely to have a larger
influence on dietary behavior than genetics.

We agree with the reviewer suggestion and change it accordingly.

Title Use of the word “awareness” is somewhat misleading as it suggests knowledge;
whereas the authors evaluated interest via Google search trends.

We understand the reviewer concerns regarding the use of the word awareness in the title;
yet it is widely used in this kind of studies and it is our perception that its use may boost
the visibility and accessibility of the work since it is common to use it. As examples we
highlight the following works:

Hu, Dingtao, et al. "More effective strategies are required to strengthen public awareness
of COVID-19: Evidence from Google Trends." Journal of global health 10.1 (2020).

Boehm, Anna, et al. "Assessing global COPD awareness with Google Trends." European
Respiratory Journal 53.6 (2019).

Lin, Ro-Ting, Yawen Cheng, and Yan-Cheng Jiang. "Exploring public awareness of
overwork prevention with big data from Google Trends: retrospective analysis." Journal
of Medical Internet Research 22.6 (2020): e18181.

Rasheed, Rajna, and Sobha Sivaprasad. "Google trends as a surrogate marker of public
awareness of diabetic retinopathy." Eye 34.6 (2020): 1010-1012
.
